# Altered Sweat Composition Due to Changes in Tight Junction Expression of Sweat Glands in Cholinergic Urticaria Patients

**DOI:** 10.3390/ijms25094658

**Published:** 2024-04-25

**Authors:** Denisa Daci, Sabine Altrichter, François Marie Grillet, Selma Dib, Ahmad Mouna, Sukashree Suresh Kumar, Dorothea Terhorst-Molawi, Marcus Maurer, Dorothee Günzel, Jörg Scheffel

**Affiliations:** 1Clinical Physiology/Nutritional Medicine, Charité—Universitätsmedizin Berlin, Corporate Member of Freie Universität Berlin and Humboldt-Universität zu Berlin, 12203 Berlin, Germany; denisa.daci@charite.de (D.D.); selma.dib@charite.de (S.D.); ahmad.mouna@charite.de (A.M.); sukashree.suresh-kumar@charite.de (S.S.K.); dorothee.guenzel@charite.de (D.G.); 2Fraunhofer Institute for Translational Medicine and Pharmacology ITMP, Allergology and Immunology, 12203 Berlin, Germanydorothea.terhorst@charite.de (D.T.-M.); marcus.maurer@charite.de (M.M.); 3Institute of Allergology, Charité—Universitätsmedizin Berlin, Corporate Member of Freie Universität Berlin and Humboldt-Universität zu Berlin, 12203 Berlin, Germany; 4Departement of Dermatology and Venerology, Kepler University Hospital, 4020 Linz, Austria; 5Center for Medical Research, Johannes Kepler University, 4021 Linz, Austria

**Keywords:** cholinergic urticaria, hypohidrosis, tight junction, paracellular pathway, claudin-10, sweat gland

## Abstract

In cholinergic urticaria (CholU), small, itchy wheals are induced by exercise or passive warming and reduced sweating has been reported. Despite the described reduced muscarinic receptor expression, sweat duct obstruction, or sweat allergy, the underlying pathomechanisms are not well understood. To gain further insights, we collected skin biopsies before and after pulse-controlled ergometry and sweat after sauna provocation from CholU patients as well as healthy controls. CholU patients displayed partially severely reduced local sweating, yet total sweat volume was unaltered. However, sweat electrolyte composition was altered, with increased K^+^ concentration in CholU patients. Formalin-fixed, paraffin-embedded biopsies were stained to explore sweat leakage and tight junction protein expression. Dermcidin staining was not found outside the sweat glands. In the secretory coils of sweat glands, the distribution of claudin-3 and -10b as well as occludin was altered, but the zonula occludens-1 location was unchanged. In all, dermcidin and tight junction protein staining suggests an intact barrier with reduced sweat production capability in CholU patients. For future studies, an ex vivo skin model for quantification of sweat secretion was established, in which sweat secretion could be pharmacologically stimulated or blocked. This ex vivo model will be used to further investigate sweat gland function in CholU patients and decipher the underlying pathomechanism(s).

## 1. Introduction

Cholinergic urticaria (CholU) is a form of chronic inducible urticaria, where patients develop itchy wheals, sometimes accompanied by angioedema, upon exposure to exercise, high environmental temperatures, and other situations that elicit sweat secretion, such as stress, hot and spicy food, hot baths, etc. [1]. The current international urticaria guideline recommends provocation testing for the diagnosis of cholinergic urticaria and a step-wise treatment approach starting with 2nd generation antihistamines and escalation with omalizumab and ciclosporin upon prior treatment failure [2]. Complete control of symptoms, however, cannot be achieved in all CholU patients, despite the use of these treatments, even in higher than licensed doses [3,4,5,6,7]. Differences in the therapy response could be due to distinct pathophysiological mechanisms that present in a similar clinical picture.

The different pathomechanisms that may contribute to the development of various subtypes of CholU have recently been reviewed by Fukunaga et al. [8,9]. The authors suggested (i) a conventional sweat allergy-type CholU; (ii) follicular-type CholU; and (iii) cholinergic urticaria with palpebral angioedema, which are believed to be formed around sweat glands due to leakage of sweat containing antigens into the dermis and IgE-mediated reactions of surrounding mast cells (types (i) and (iii)) or via serum factor and acetylcholine stimulation of mast cells around the hair follicles to form wheals consistent with the hair follicles (ii). A fourth type (iv) of cholinergic urticaria with acquired anhidrosis and/or hypohidrosis (CholU-Anhd) and reduced cholinergic receptor muscarinic 3 (CHRM3) expression had been demonstrated by colleagues [9,10] and us [11]. It was proposed that the reduced receptor expression on sweat glands leads to reduced sweat production of the sweat gland and the non-uptake and degradation to an acetylcholine overflow that could promote degranulation from the adjacent mast cells [12]. A patchy distribution of hypohidrotic and normohydrotic areas in these patients had been described [10]. Furthermore, poral occlusion has been proposed to be involved in the induction of sweat leakage into the surrounding tissue [13] and thus contribute to CholU development.

Reduced sweat volume due to leakage of sweat into the tissue surrounding the sweat glands has been observed in the affected skin of atopic dermatitis patients [14]. In the sweat glands of these patients, the paracellular barrier was impaired. Paracellular barrier properties are determined by the composition of the tight junction (TJ), a multi-molecular structure that assembles into a TJ strand network between two neighboring epithelial cells. The main constituents of the TJ strands are members of the claudin (CLDN) protein family, small (~22–26 kDa) tetraspan membrane proteins which interact with their counterparts from the neighboring cells through their two extracellular segments (for review, see e.g., [15,16]). Through their C-terminal PDZ-binding motif, they interact with cytoplasmic scaffolding proteins, such as zonula occludens 1 (ZO-1). Whereas some claudins seal the paracellular pathway (barrier-forming claudins, e.g., CLDN1 and CLDN3), others (e.g., CLDN2, CLDN10, CLDN15) convey a specific charge selectivity to the paracellular pathway as they are forming charge- and size-selective paracellular channels by means of their extracellular segments. A further TJ protein, occludin (OCLN), is an integral constituent of most TJ strands. Its function is less understood; however, it appears to contribute to the sealing of the paracellular barrier [17].

In sweat glands, TJ strand composition differs between the secretory coil (CLDN3, CLDN4, CLDN10b, OCLN) and the sweat duct (CLDN1, CLDN3, CLDN4, CLDN15, OCLN) [14]. In the secretory coil, CLDN3 and OCLN are predominantly localized to the TJs of the central duct, whereas CLDN10b is almost exclusively found in the TJs of the canaliculi branching off this central duct [18,19,20]. ZO-1 is found to be part of all sweat gland TJs.

In atopic dermatitis patients, reduced sweat secretion was found to be due to a downregulation of the major barrier-forming TJ proteins in the sweat duct (CLDN1 and CLDN3) and in the secretory coil (CLDN3) [14]. This loss of paracellular barrier function was visualized by dermcidin staining that documented the presence of dermcidin in the tissue surrounding the sweat glands rather than being restricted to the sweat gland lumen [14,21].

Hypohidrosis or even anhidrosis is found in patients suffering from HELIX syndrome, a complex hereditary disease caused by bi-allelic mutations in the *CLDN10* gene [18,19,20,22,23,24,25]. CLDN10b, one of the two major CLDN10 isoforms [26,27] encoded by *CLDN10*, forms a paracellular cation channel. In eccrine sweat glands, this isoform is specifically expressed in the secretory coil and localized in the canaliculi [18], where it appears to facilitate Na^+^ secretion into the lumen, driven by the lumen-negative potential generated by active, transcellular Cl^−^ secretion. The resulting luminal NaCl accumulation gives rise to the osmotic gradient necessary for water movement into the sweat gland lumen [28].

Thus, barrier loss as well as loss of the specific cation permeability of the paracellular pathway are potential causes for a reduction in sweat volume. It was, therefore, the aim of the present study to investigate whether a dysregulation of TJ proteins and the resulting impairment of the paracellular properties may contribute to the reduced sweat production in CholU patients. We further aimed at establishing an ex vivo skin model to study potential regulators of sweating under defined conditions. In the future, such a model may help to directly correlate alterations in sweat production with sweat gland histology.

## 2. Results

### 2.1. Reduced Sweating Can Be Observed in a Subgroup of CholU Patients

In CholU patients who underwent pulse-controlled ergometry (PCE), a subgroup showed reduced (7/13, 54%) or severely reduced localized sweating (3/13, 23%), but none of the healthy controls [11]. Severely reduced sweating was associated with significantly higher disease severity, reduced acetylcholine receptor CHRM3, and cholinesterase expression of the sweat glands in these CholU patients [11].

### 2.2. No Sweat Leakage from Sweat Glands in CholU Patients with Reduced Sweating

To test the hypothesis that local sweat production is decreased in CholU patients because a barrier defect allows primary sweat to diffuse from the gland into the interstitium of the surrounding tissue, dermcidin staining was performed on skin biopsies of seven CholU patients and five healthy controls after PCE. The presence of dermcidin within the sweat glands was confirmed for both groups; however, no dermcidin signals in the tissue surrounding the sweat glands were detected (Figure 1).

### 2.3. Altered TJ Protein Distribution in Sweat Gland Secretory Coils from CholU Patients

To investigate TJ integrity and localization in sweat gland secretory coils of CholU patients, different tight junction components were stained and compared with healthy controls. The barrier-forming TJ protein CLDN3 was strongly expressed and localized in the TJs of the central ducts (arrowheads in Figure 2a) of the secretory coils. In healthy controls, CLDN3 was almost absent from the canaliculi extending from these ducts that were clearly stained by the cytoplasmic TJ-associated scaffolding protein ZO-1 (arrows in Figure 2a), however, canalicular CLDN3 staining was more intense in sweat glands from CholU patients (arrows in Figure 2a). A similar pattern was found for the TJ protein OCLN (Figure 2b), indicating, that the barrier function of the sweat glands was intact.

In contrast, the paracellular cation channel-forming CLDN10b was exclusively localized in the canaliculi of the secretory coils of healthy control sweat glands (Figure 2b,c). In sweat glands from affected skin regions of CholU patients, this pattern was distinctly altered and CLDN10b staining was reduced in the canaliculi (arrows in Figure 2b,c), but spread into the TJs of the central duct (arrowheads in Figure 2b,c).

Staining for the cytoplasmic scaffolding protein ZO-1 was positive in both junctional compartments (arrows and arrowheads in Figure 2a,c) of the secretory coils from healthy controls and appeared essentially unaltered in both the central duct and the canalicular TJs in samples from CholU patients (Figure 2a,c).

Images were analyzed using the co-localization tool of the Zeiss ZEN 3.0 SR (black) software (version 16.0.1.306) and results are shown in Figure 2d–f. ZO-1 signals were used as TJ markers and the percentage of ZO-1 colocalized with CLDN3 or CLDN10 was evaluated. Whereas the percentage of CLDN3-positive TJs was not affected in CholU patients (Figure 2d), the percentage of CLDN10-positive TJs decreased, indicating a general loss of CLDN10 in CholU patient sweat glands (Figure 2f). In contrast, the percentage of OCLN-positive signals co-localizing with CLDN10 was increased in CholU patient sweat glands, indicating that the redistribution of OCLN towards the canaliculi and of CLDN10 from the canaliculi towards the central duct results in a stronger overlap of the two proteins (Figure 2b,e).

### 2.4. Altered Electrolyte Composition of Sweat from CholU Patients Compared to Healthy Controls

To assess the sweat volume and electrolyte composition, sweat was collected in a cohort of 39 CholU patients and 59 healthy controls during a 15 min sauna visit. Demographic data as well as collected sweat volume are listed in Table 1.

Sweat volume differed between male and female participants within both groups (Figure 3); however, there was no significant difference in total sweat volume between CholU patients and healthy controls, neither in the combined data, nor when comparing only males or females.

As CLDN10b provides the main pathway for Na^+^ secretion in the secretory coil [18], we hypothesized that sweat electrolyte composition might be altered in affected skin of CholU patients and that these differences might be large enough to be detectable in whole-body sweat. We therefore assessed the pH and sweat electrolyte concentrations in the collected sweat samples. No significant differences were found in sweat pH, Na^+^, Cl^−^, and Mg^2+^ concentrations (Table 2). However, sweat K^+^ and Ca^2+^ concentrations were increased in sweat from CholU patients compared to healthy controls.

Furthermore, K^+^/Cl^−^ ratios were increased in sweat from CholU patients compared to healthy controls (Figure 4), indicating alterations in the underlying secretory processes.

### 2.5. Establishing a Model for Sweat Provocation in Isolated Skin

In an attempt to develop a model to investigate sweat production ex vivo, skin pieces from plastic surgery from healthy controls were flattened out and fixed with needles on a Sylgard silicon-filled dissection tray. Adjacent patches of about 2 cm × 5 cm were brushed with iodide tincture and, after drying, covered with a mixture of 10% (weight/volume) starch with oil. In 22 out of 28 skin samples, baseline sweat production was detectable as small black punctate sweat accumulations on the day of surgery (Figure 5).

Sweat production could be stimulated by intradermal carbachol injection (Figure 6a) and inhibited by injection of tetrodotoxin (TTX) (Figure 6b). Carbachol-stimulated sweat production could be reduced if deodorant (active component: aluminum chlorohydrate) was applied to the skin prior to carbachol injection (Figure 6c), further indicating that the observed reaction was due to sweat secretion rather than unspecific fluid accumulation.

Alterations in sweat secretion induced by all three treatments were due to both the number of secreting sweat glands and the amount of sweat secreted per gland (average spot size). However, especially for carbachol and TTX, the effect on the number of secreting sweat glands was more pronounced. Thus, in the presence of carbachol, the median number of active sweat glands observed 15 min after adding the oil-starch mixture to the skin was more than twice that of the unstimulated skin area (median 211% of baseline, range 152–280%, *n* = 11 pairs of test and control areas from 4 different skin samples), after TTX application the number of active sweat glands observed 15 min after adding the oil-starch mixture was only 64% of that observed in untreated skin (range 43–64%, *n* = 8 pairs of test and control areas from 3 different skin samples). Corresponding average spot areas amounted to a median value of 132% for carbachol (range 126–199%) and 95% for TTX (range 56–97%). Application of deodorant to carbachol-stimulated skin areas affected both parameters more similarly; 15 min after adding the oil-starch mixture, the median number of active sweat glands was 68% of control (range 48–86%) and the median average spot size 76% of control (range 60–142%; *n* = 12 pairs of test and control areas from 6 different skin samples).

In order to test whether sweat glands could be kept functional over a longer period of time, skin patches were cut from fresh skin samples and cultured for up to 14 days. For a duration of up to 7 days, immunofluorescence staining of TJ proteins looked as described above for fresh biopsies from healthy controls. However, despite their normal appearance, no baseline sweat production or carbachol-induced sweat production could be observed, even after only 24 h in culture. Interestingly, at cultivation periods longer than 7 days, CLDN10 and OCLN signals started to be mislocalized in a similar way as observed in CholU patients (Figure 7).

## 3. Discussion

To our knowledge, this study is the first to investigate dysregulations of the TJ in sweat glands of CholU patients. TJs are complex structures that seal the paracellular cleft against unwanted passage of solutes. However, depending on the claudins expressed in an epithelium, TJs may specifically allow paracellular transport of certain ions [26,29,30]. One of these channel-forming claudins is the cation channel-forming CLDN10b, biallelic mutations of which cause hypohidrosis or even anhidrosis in affected patients [25]. Previous studies from Asian patient populations describe CholU with anhidrosis and/or hypohidrosis (CholU-Anhd), where affected patients suffer from generalized hypo- or anhidrosis and sweat prick tests are negative [9]. In our cohort, we also identified patients with hypohidrosis, showing reduced sweating in the affected areas and, in part, even hyperhidrosis in unaffected skin areas like the face or armpits. In one study on CholU patients, a patchy distribution of hidrotic and hypohidrotic areas was also clearly depicted [13].

In our study cohort that was exposed to heat in a sauna, we observed differences between sweat volumes between male and female participants within both groups, healthy controls and CholU patients (Figure 3). These differences may be due to various reasons such as differences in body surface, total body water, smaller sweat glands with reduced sensitivity to cholinergic stimuli, etc. [31,32]. However, there was no significant difference in total sweat volume between CholU patients and healthy controls, neither in the combined data nor when comparing only males or females. One reason for this lack of effect may be the patchy distribution of areas with hypohidrosis and areas with compensatory hyperhidrosis. In conclusion, the finding that overall sweat volume obtained during sauna sweat provocation did not differ between CholU patients and healthy controls argues against a generalized hypo-/anhidrosis in our study cohort.

Local reduction in sweating may be caused by many factors: sweat leakage into the surrounding tissue due to pore blocking [9,33] or sweat gland barrier dysfunction by downregulation of barrier-forming tight junction proteins, as demonstrated in sweat glands of atopic dermatitis patients [14]. Both pore blocking and loss of barrier-forming TJ proteins appeared not to be the case in the studied CholU individuals. On the one hand, there was no leakage of dermcidin, a common sweat component, into the dermal tissue. On the other hand, there was no evidence for a downregulation of the barrier-forming tight junction protein CLDN3 [34,35] or of OCLN, the function of which is not yet fully solved but has also been associated with barrier formation [17,36,37]. Both CLDN3 and OCLN appeared unaltered in the central duct of sweat glands from CholU patients and both even spread into the canaliculi of the secretory coils from which they are excluded in sweat glands from healthy controls, suggesting an intact barrier preventing sweat leakage.

For the cation channel-forming CLDN10b, we found a reciprocal distribution. In healthy controls, CLDN10b localization is restricted to the canaliculi of the secretory coils but absent from TJs of the central duct. This pattern changes distinctly in CholU patients, where CLDN10b was reduced or even lost from the canaliculi, where it was replaced by CLDN3 and OCLN.

This replacement of CLDN10b by CLDN3 is reminiscent of the situation in HELIX syndrome patients. In the sweat glands of these patients, CLDN3 and OCLN are also found in the canaliculi devoid of CLDN10b. Apart from dysfunction of various glands, HELIX syndrome is also characterized by electrolyte imbalance, caused by CLDN10b deficiency in the thick ascending limb of Henle’s loop (TAL) in the nephron. Similar to the situation in sweat glands, CLDN3 and CLDN10b are both expressed in the TAL but are found exclusively in different TJ segments in a mosaic-like pattern [38,39]. In the absence of CLDN10b from these segments, e.g., in CLDN10b knockout mice and in TAL from a HELIX syndrome patient, CLDN3 (together with two further claudins, CLDN-16 and -19) is present in all TJ segments, i.e., also in those that are normally occupied by CLDN10b [19,24,38].

Unlike the situation in HELIX syndrome patients, however, in CholU patients CLDN10b spread into the TJs of the central duct. This mislocalization will have two consequences:

(i) It will render the paracellular pathway of the canaliculi less permeable to monovalent cations, especially to Na^+^. As depicted in Figure 8, the paracellular Na^+^ permeability is essential for sweat formation. Sweat formation is driven by active, transcellular Cl^−^ secretion. Due to the basolateral localization of the Na^+^/K^+^ ATPase, Na^+^ cannot be secreted along the transcellular route. CLDN10b provides the basis for paracellular Na^+^ secretion, driven by the lumen-negative potential. Together, Na^+^ and Cl^−^ increase the osmolarity of the primary sweat, so that water can follow through Aqp5 water channels (for review, see e.g., [28]). The loss of the paracellular Na^+^ transport pathways due to CLDN10b downregulation/redistribution may be compensated by transcellular K^+^ secretion through apical K^+^ channels [28] due to an increased lumen-negative potential (Figure 8). This should cause an increase in the K^+^ content of the sweat, similar to the increased K^+^ content in the saliva observed in a HELIX syndrome patient [20].

(ii) It will render the paracellular pathway of the central duct in the secretory coil more permeable to monovalent cations, especially to K^+^. If indeed the K^+^ concentration in the primary sweat is increased, K^+^ may leak back into the surrounding tissue along its concentration gradient and may increase local interstitial K^+^ concentrations, which, e.g., may sensitize sensory nerve fibers and contribute to the itch/pain sensation experienced by CholU patients.

Sweat electrolyte concentrations in healthy controls were in the same range as published data [40,41]. In CholU patients, however, sweat K^+^ and Ca^2+^ concentrations were increased, supporting the outlined scenario. The mechanism for the Ca^2+^ increase is unclear. However, the increased K^+^/Cl^−^ ratio further indicates that the loss in Na^+^ secretion is in part compensated by increased K^+^ secretion. As the sweat was collected from the whole body, i.e., from both affected and non-affected skin areas, it is not surprising that Na^+^ concentrations did not differ significantly between CholU patients and healthy controls in the present study. Furthermore, Na^+^ and Cl^−^ are reabsorbed along the sweat duct along highly specific pathways involving the epithelial Na^+^ channel ENaC and the CFTR Cl^−^ channel [42]. This reabsorption may mask alteration in Na^+^ content in the primary sweat.

Studies on human sweat glands and their function are restricted by limited in vitro models that used challenging culture methods [43] or used animal models on primates [44] or sweating of footpaths of mice [45], which probably do not completely resemble typical human eccrine sweat glands on the body surface. To overcome these challenges and to build a base for further studies, we developed an ex vivo skin model. Ex vivo skin samples showed physiological basal sweating functions that could be stimulated by carbachol or inhibited by TTX or application of aluminum chlorohydrate-containing deodorant. Ex vivo culture showed stable TJ protein localization for up to 7 days. However, sweat secretion already stopped after day one, i.e., long before CLDN10b mislocalization started. A possible hypothesis is that reduced cholinergic stimulation of the sweat glands, due to lacking neuronal input in skin explants, or due to the receptor downregulation observed in CholU patients [10,11], causes CLDN10b mislocalization in the secretory coil of eccrine sweat glands.

A limitation of our study is the overall low number of patients included. Moreover, patients who come to our tertiary center typically have severe disease and might not represent the average patient population. In addition, the analysis of sweat content was only performed from the whole sweat and not from different regions where sweat reduction was observed. Finally, our skin model used skin from plastic surgery procedures that were not performed in our institute. Therefore, there were differences between surgery and the start of the experiment of up to several hours, which may have affected the overall performance of the assay.

In summary, despite the observed differences in TJ composition and distribution in CholU patients and the changes in the cation composition of the sweat, the present study does not allow us to conclude that CLDN10b mislocalization is the primary reason for the reduced sweat secretion in CholU patients. The mechanisms causing alterations in TJ protein localization are yet vastly unknown. All findings may be triggered by the same yet unidentified stimulus, or may, indeed, even be unrelated to each other. Further studies are needed to elucidate a possible causality between these observations in this complex disease.

## 4. Materials and Methods

### 4.1. Cohorts of CholU Patients and Healthy Controls

Thirty-nine patients diagnosed with CholU by pulse-controlled ergometry (PCE [46]) were recruited at the Urticaria Centre of Reference and Excellence (UCARE) of the Department of Dermatology and Allergy, Charité—Universitätsmedizin, Berlin, for provocation testing and extensive clinical characterization. Fifty-nine healthy controls balanced for both sexes also participated in the study. All participants were asked to avoid taking H_1_-antihistamines for at least 3 days and local or systemic steroids or other immunosuppressive therapy for 2 weeks before skin testing. Also, it was ascertained that no exercise or any other factor-induced crisis had happened during the last 24 h before testing. All individuals underwent sauna provocation. Demographic and clinical characteristics of the patients included are shown in Table 1. This study was approved by the Ethics Committee of the Charité—Universitätsmedizin Berlin (#EA4/124/10) and registered with the German Clinical Trials Registry (DRKS-ID: DRKS00004277). Part of the study data had been published elsewhere [47].

Additionally, 13 patients underwent PCE, sweating assessment using Minor’s test [48,49], and 5 mm skin biopsy before and after PCE. After informed consent, patients were advised to stop antihistamine intake at least 3 days before any of the mentioned tests below. None of the patients had been taking local or systemic steroids or other immunosuppressive therapy in the last 2 weeks before the tests. Twelve matched healthy volunteers who underwent the same procedures served as the control group. The demographic characteristics of the study participants have previously been published [11]. This study was approved by the Ethics Committee of the Charité—Universitätsmedizin Berlin (#EA1/241/15) and registered in the German Clinical Trials Register (DRKS-ID: DRKS00012755). Part of the study data had been published elsewhere [11].

For the ex vivo skin model, abdominal skin explants obtained from plastic surgery procedures were used after informed consent of the skin donor was provided. The study was approved by the Ethics Committee of the Charité—Universitätsmedizin Berlin (#EA4/099/21).

### 4.2. Sweat Collection in CholU Patients and Healthy Controls

Patients and healthy controls undergoing a sauna provocation test were placed at 80 °C for 15 min inside a plastic bag (polyethylene, food-safe grade, Ratioform GmbH, Pliening, Germany), after showering without soap and complete drying. All body parts except the head were covered by the bag, whole-body sweat was collected in the bag, and total sweat volume was assessed.

### 4.3. Immunofluorescence Staining

Tissue from skin biopsies was fixed in 4% formalin (PFA, Carl Roth, Karlsruhe, Germany), dehydrated in an increasing ethanol series, embedded in paraffin, and cut into 5 µm sections using a rotary microtome (pFM Rotary 3000 Compact, pfm medical GmbH Köln, Germany).

Directly prior to immunofluorescence staining, Tris-EDTA-citrate (TEC) buffer was prepared by diluting the stock solution 1:20 in distilled water (1 L TEC stock solution: Tris(hydroxymethyl)aminomethane, 5 g; disodium ethylenediaminetetraacetate dihydrate, 10 g; trisodium citrate, 6.4 g; dissolved in distilled water, pH adjusted to 7.8). Sections were deparaffinized in xylene and rehydrated in an ethanol series (100%, 90%, 80%, 70%, distilled water). For antigen retrieval, sections were microwaved in TEC buffer (360 W, 3 × 10 min with brief intervals to refill TEC buffer if necessary) and subsequently incubated in a 0.001% trypsin solution for 10 min at 37 °C. After rinsing in Tris Buffered Saline with 0.1% Tween 20 (TBST), sections were blocked in Dulbecco’s Phosphate buffered solution with CaCl_2_ and MgCl_2_ (PBS+, Gibco, Darmstadt, Germany) with 4% goat serum at room temperature for 30 min. Primary antibodies were diluted 1:100 (CLDN antibodies) or 1:200 (ZO-1 and OCLN antibodies) in blocking solution and sections incubated overnight at 4 °C (rb anti-CLDN3, #34-1700, Thermo Fisher/Invitrogen, Waltham, MA, USA; rabbit anti-CLDN10, #38-8400, Thermo Fisher/Invitrogen; m anti-OCLN, #33-1500, Thermo Fisher/Invitrogen; m anti-ZO-1, 610966, BD Transduction Laboratories, Heidelberg, Germany). After 3 × 5 min rinsing in TBST, secondary antibodies were diluted 1:400 in blocking solution (goat anti-mouse Alexa 488, goat anti rabbit Alexa 488, goat anti-mouse Alexa 594, goat anti-rabbit Alexa 594, A32723, A32731, A32742, A32740, Thermo Fisher; Cy2 IgG goat anti-rabbit, Cy5 IgG goat anti-mouse, 111-225-144, 111-175-144, Jackson ImmunoResearch Laboratories, Ely, UK), 4′,6-diamidino-2-phenylindole dihydrochloride (DAPI, Roche AG, Mannheim, Germany) was added at 1:1000, and sections were incubated for 60 min at room temperature. Sections were then rinsed 1 × 5 min in TBST, 1 × 5 min in PBS+, 2 × 3 min in distilled water, dehydrated in 100% ethanol, and embedded in ProTaqs Mount Fluor (Biocyc, Luckenwalde, Germany).

For dermcidin staining, 4 µm sections were deparaffinized in xylene and rehydrated in an ethanol series (100%, 90%, 80%, 70%, distilled water). For antigen retrieval, sections were incubated in citrate buffer pH 6.0 (#S2369, Dako/Agilent, Santa Clara, CA, USA) in a steamer at 120 °C for 11 min, rinsed with PBS^+^, and blocked with serum-free protein lock solution (#X0909, Dako) for 30 min at room temperature. Anti-dermcidin primary antibody (#sc-33656, clone G-81, SantaCruz, Dallas, TX, USA) was diluted 1:50 in antibody diluent (#S3022, Dako) and incubated overnight at 4 °C. After 3 × 5 min rinsing in TBST, secondary antibodies were diluted 1:400 in blocking solution (goat anti-mouse Alexa 488), DAPI was added at 1:400, and sections were incubated for 60 min at room temperature. Sections were then rinsed 3 × 5 min in distilled water, dehydrated in 100% ethanol, and embedded in ProTaqs Mount Fluor (Biocyc, Luckenwalde, Germany).

Imaging was performed on a Zeiss LSM 780 confocal microscope (Zeiss, Jena, Germany).

Images were analyzed using the co-localization tool of the Zeiss ZEN 3.0 SR (black) software (version 16.0.1.306). Thresholds were set so that the TJ signals were clearly positive. This resulted in four categories of pixels: (1) high for protein 1 but low for protein 2, (2) high for protein 2 but low for protein 1, (3) high for protein 1 and protein 2, (4) low for protein 1 and protein 2. With protein 1 being the TJ marker ZO-1 (or OCLN, respectively) and protein 2 being CLDN3 or CLDN10, respectively, the percentage of claudin-positive TJs was calculated as: (number of pixels in category 3)/(number of pixels in category 1 + 3).

For more detailed information see Appendix A.

### 4.4. pH and Electrolyte Measurements

Sweat pH was measured with a standard pH meter (AE150, Fisher Scientific, Schwerte, Germany). Sweat electrolyte concentrations were determined in a Stat Profile Prime^®^ Electrolyte Analyzer (Nova Biomedical, Mörfelden-Walldorf, Germany).

### 4.5. Ex Vivo Skin Model for Sweat Provocation and Inhibition

Skin pieces obtained after plastic surgery, with an approximate area of 10 cm × 5 cm, were stretched out and fixed on a Sylgard silicone-filled dissection tray using injection needles (Becton Dickinson, Drogheda, Ireland). The skin was cleaned and freed of any substances that could interfere with the test. At room temperature (~25 °C), a small amount of the iodine solution (Lugol-Solution: iodine 2.0% (weight/volume), potassium iodide 3.0% (weight/volume), in distilled water) was applied to two adjacent skin areas of about 1 cm × 2.5 cm each, using a cotton swab. After letting it dry, a mixture of potato starch with sunflower oil (10% weight/volume, [14]) was applied over the iodine solution. The appearance of dark blue-black dots on the tested skin indicated positive sweat production. To stimulate sweat production, 10 µL of a 100 µM carbachol solution (stock solution of 10 mM carbachol (Sigma-Aldrich, Taufkirchen, Germany) in dimethyl sulfoxide (DMSO, Carl Roth), diluted 1:100 in PBS^+^, was injected intradermally into one of the skin patches, using a Hamilton Microliter™ syringe. Alternatively, sweat secretion was inhibited by the injection of 10 µL of 10 µM tetrodotoxin (TTX, RBI/Merck, Darmstadt, Germany, stock solution of 1 mM TTX in 5 mM Na_3_-citrate, diluted 1:100 in PBS^+^). Injected patches were compared to the adjacent control patch, which was injected with 10 µL DMSO, diluted 1:100 in 5 mL PBS^+^. Furthermore, the effect of deodorant application on carbachol-stimulated sweat production was tested. In this case, deodorant (Nivea “Anti-Transpirant Invisible 48 h”; active component: aluminum chlorohydrate; Beiersdorf AG, Hamburg, Germany) was applied to one of the two adjacent skin test areas with a cotton swab before the application of iodide. Both patches were then injected with carbachol to induce sweat formation.

### 4.6. Evaluation of Sweat Production

ImageJ (Version 2.9.0) is a freely available image processing tool. As it is effective at identifying and counting individual particles in scanned images, it was employed to analyze sweat secretion in the present study. The analysis involved setting lower and upper limits for the pixel area, which determines the minimum and maximum size of dots to be counted. Due to variations between different tissues, limits were set individually, so that all sweat glands were accurately counted. The software outputs a count of the dots in the image, which corresponds to the number of active glands for that sample (as shown in Figure 5). This count was divided by the area of treated skin to obtain the active sweat glands per unit area.

The following steps were performed: (1) Loading the photographed image containing a control and test area in the ImageJ (File–Open), displaying an RGB image. (2) Two Regions of Interest (ROI) of identical shape and size were selected, one for the control and one for the test area, and saved as separate images. (3) The image type was set to 8-bit grayscale (Image toolbar–Type–8-bit). (4) Maximum and minimum threshold were set to each image, until the selected areas overlapped the maximum sweat signals detected in the original photograph, before converting the image to a binary (black and white) image (Process toolbar–Binary–Make binary; Image–Adjust–Threshold–default/Iso_data–set). When the secreted sweat dots were in groups, making it difficult to set the threshold manually to default, automatic thresholding (Iso_data) was used. (5) The image was converted into binary and converted into a mask (black dots on a white background, see Figure 5). If dots were not fully separated, a watershed algorithm was used to distinctly separate the dots (Process–Binary–Watershed). (6) The processed image was analyzed (Analyze–Analyze Particles) and two outputs were generated: summary, displaying the count, total area, average size, percentage area; results, displaying the area of each dot.

### 4.7. Skin Explant Culture

In order to culture skin explants for further experiments, a 6 mm diameter biopsy punch (Kai Medical, Solingen, Germany) was used to take several biopsies from the larger skin tissues obtained shortly after plastic surgery. These biopsies were sterilized using cotton pads moistened with 80% ethanol (denatured; Th. Geyer GmbH & Co. KG, Renningen, Germany) in order to eliminate any potential contaminants that could interfere with the culture and excess fat was removed. The biopsies were then placed in a sterile 12-well culture dish (Sarstedt, Nümbrecht, Germany) containing Dulbecco’s modified Eagle’s medium (DMEM, Gibco), supplemented with 10% fetal bovine serum (Sigma-Aldrich), 1% anti-anti (10.000 units penicillin, 10 mg streptomycin, and 25 μg amphotericin B, Sigma-Aldrich) and 1% L-glutamine (Sigma-Aldrich). The culture dishes were placed in an incubator (humidified atmosphere of 5% CO_2_, 37 °C). The medium was changed every 2 to 3 days and single skin explants were removed every 2 days, fixed overnight with 4% paraformaldehyde (Carl Roth) for further processing, and stained as described above.

### 4.8. Statistics

Unless otherwise stated, all values are given as means ± SEM. For statistical analyses, unpaired, two-tailed Student’s *t*-tests were performed.

## Figures and Tables

**Figure 1 ijms-25-04658-f001:**
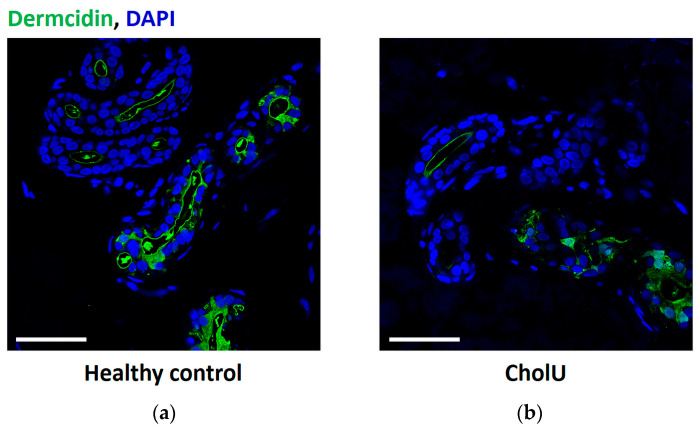
Dermcidin staining (green) in sweat glands from (**a**) healthy controls and (**b**) CholU patients shows clear signals within the sweat glands but not in the surrounding tissue. Images are representatives from *n* = 7 (CholU) and 5 healthy controls (HC). Scale bars: 50 µm.

**Figure 2 ijms-25-04658-f002:**
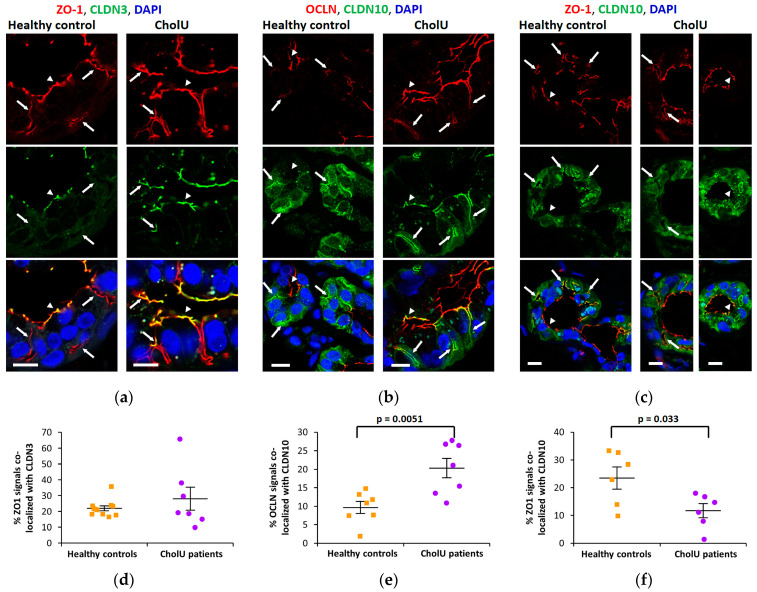
Representative fluorescence staining of TJ proteins in sweat glands from healthy controls and CholU patients. (**a**) Staining for ZO-1 (red) and CLDN3 (green); (**b**) Staining for OCLN (red) and CLDN10 (green); (**c**) Staining for ZO-1 (red) and CLDN10 (green). Arrows indicate canalicular TJs, arrowheads TJs of the central duct. Scale bars, 10 µm. (**d**–**f**) Evaluation of the co-localization of the TJ markers ZO-1 and OCLN with CLDN3 and CLDN10 signals, respectively. (**d**) The percentage of ZO-1-positive structures positive for CLDN3 did not differ between healthy controls (34 images from 11 healthy controls) and CholU patients (25 images from 7 CholU patients), indicating that the total amount of CLDN3 was unchanged. (**e**) The percentage of OCLN-positive structures positive for CLDN10 was larger in CholU patient samples (22 images from 7 CholU patients) compared to samples from healthy controls (12 images from 7 healthy controls), indicating that the redistribution of OCLN and CLDN10 within the TJ results in a stronger overlap of the two proteins. (**f**) The decrease in the percentage of ZO-1-positive structures positive for CLDN10 in CholU patient samples (18 images from 6 CholU patients) compared to samples from healthy controls (23 images from 6 healthy controls) indicates a general loss of CLDN10 from CholU patient TJs.

**Figure 3 ijms-25-04658-f003:**
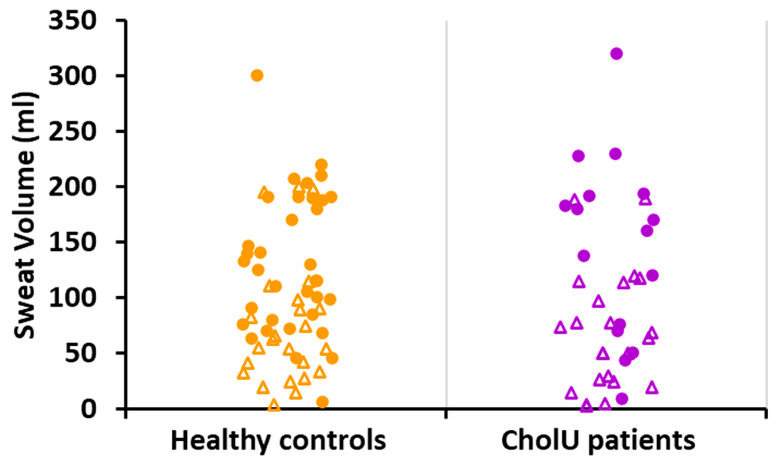
Whole-body sauna-induced total sweat volumes from healthy controls and CholU patients (circles, males; triangles, females). In both groups, average volumes were larger in males compared to females; however, there were no significant differences between healthy controls and CholU patients in either group (for statistics, see Table 1).

**Figure 4 ijms-25-04658-f004:**
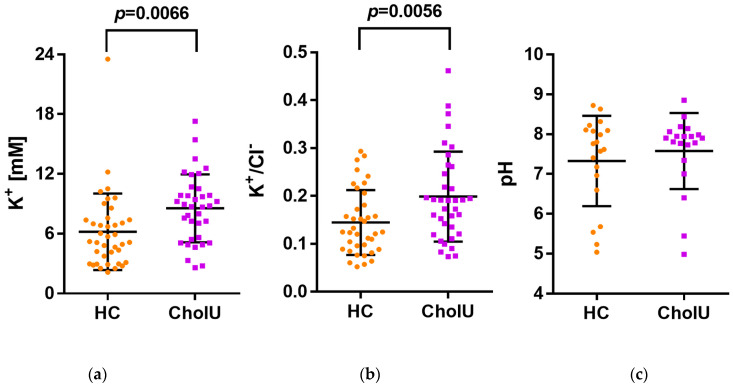
CholU patients displayed significantly elevated concentrations of (**a**) K^+^ as well as an increased (**b**) K^+^/Cl^−^ ratio in their sweat compared to healthy controls (HC), while the (**c**) pH was not altered. Data presented as mean ± SD.

**Figure 5 ijms-25-04658-f005:**
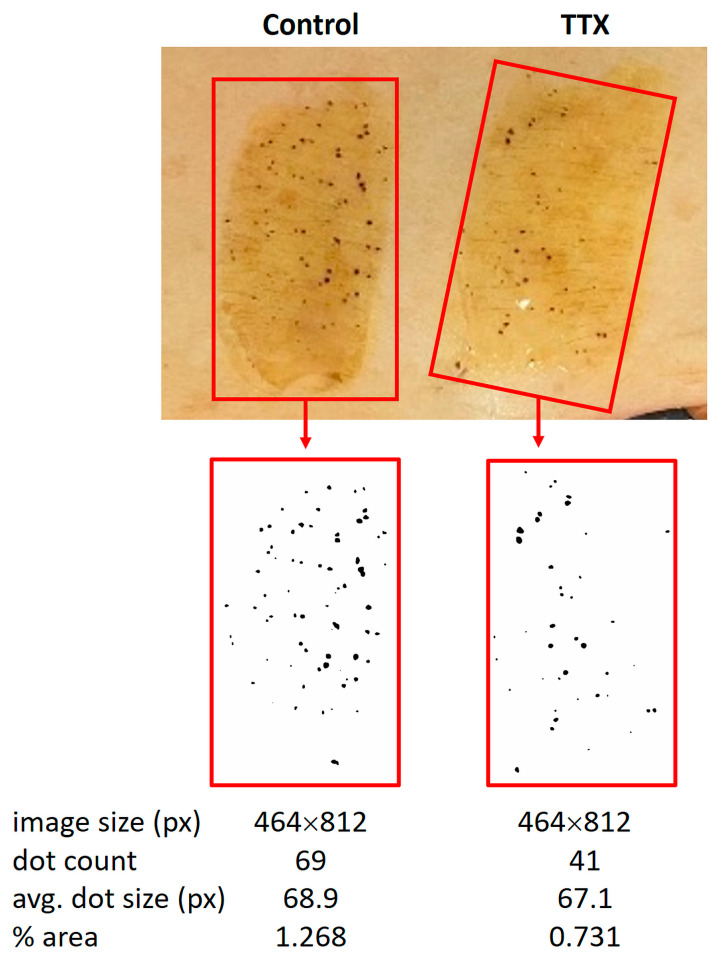
Quantification procedure of sweat production in isolated skin. In this example, control and TTX-injected skin areas were evaluated for sweat secretion. Sweat secretion was visualized by an iodine/starch reaction under oil in isolated skin. Secreted sweat could be identified as black dots that were photographed every 5 to 10 min. ImageJ (Version 2.9.0) was employed to convert the photographs into black and white images to evaluate the number and average (avg.) area of these dots (unit: number of pixels, px). The % area covered by sweat was calculated within each given experimental area (red rectangle). A summary of the quantifications is shown in Figure 6.

**Figure 6 ijms-25-04658-f006:**
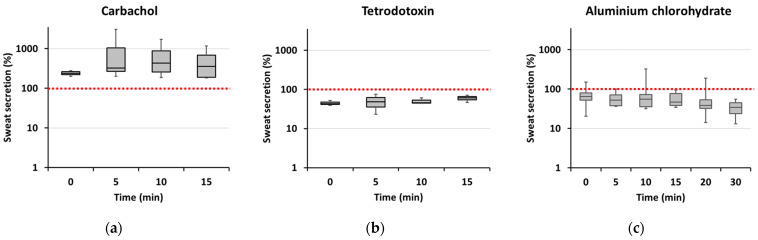
Time course of sweat secretion in isolated skin samples on the day of surgery. Quantification was carried out as exemplified in Figure 5. (**a**) Carbachol-stimulated sweat secretion (% of baseline secretion, *n* = 4 to 11 from 4 different patients for individual time points); (**b**) Inhibitory effect of TTX on sweat secretion (% of baseline secretion; *n* = 3 to 8 from 3 different patients for individual time points); (**c**) Inhibitory effect of local deodorant application (active component: aluminum chlorohydrate) on carbachol-induced sweat secretion (% of carbachol-induced sweat secretion in the absence of deodorant; *n* = 7 to 18 from 15 different patients for individual time points). Box plots depict the median, 1st and 3rd quartiles; whiskers indicate maximum and minimum values for each time point.

**Figure 7 ijms-25-04658-f007:**
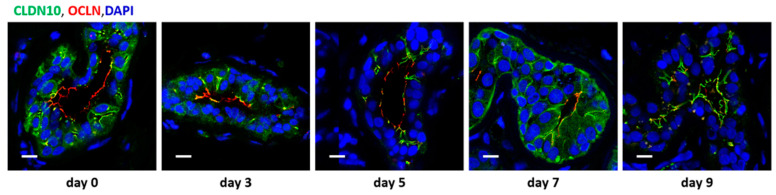
Immunofluorescence staining of the tight junction proteins CLDN10 (green) and OCLN (red) in sweat glands from healthy cultivated skin samples. From day 0 to day 5 of tissue culture, CLDN10 is localized in the canaliculi, OCLN in the central duct. On day 7 of tissue culture, the first signs of CLDN10 and OCLN signal overlap (yellow signal) become visible. On day 9 of tissue culture, signals of CLDN10 and OCLN are clearly overlapping. Scale bars, 10 µm. Representatives of 3 to 8 images per time point from 4 different patients.

**Figure 8 ijms-25-04658-f008:**
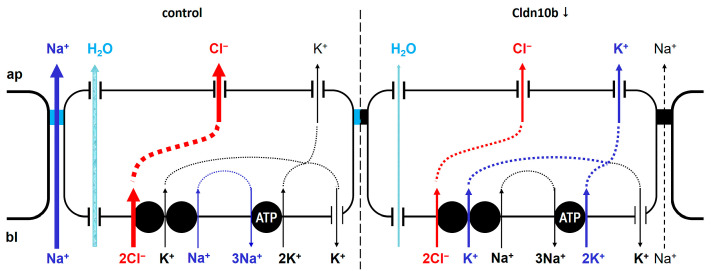
Schematic of the transport processes involved in sweat secretion in the presence and absence of CLDN10b. Under control conditions in the presence of CLDN10b, Cl^−^ is actively secreted via the basolateral Na^+^-K^+^-2Cl^−^ symporter (NKCC1) and apical Cl^−^ channels. Na^+^ is removed from the cytoplasm via the Na^+^/K^+^ ATPase, and K^+^ is extruded via basolateral K^+^ channels [28]. Cl^−^ secretion generates a lumen-negative transepithelial potential that drives paracellular Na^+^ secretion through CLDN10b-based paracellular channels. Water follows along the developing osmotic gradient through aquaporin-5 channels. Loss of CLDN10b will prevent paracellular Na^+^ secretion and thus increase the lumen-negative potential. This, in turn, will favor K^+^ secretion through apical K^+^ channels, so that at least some basal production of K^+^-rich sweat may be maintained.

**Table 1 ijms-25-04658-t001:** Participants in sauna sweat collection.

	Healthy Control	CholU Patient	*p*-Value (*t*-Test)
Number (m/f)	59 (31, 28)	39 (16, 23)	
Age ([years]; mean ± SEM; *n*)	30.76 ± 0.99; 59	34.74 ± 1.93; 39	0.05
BMI ([kg/m^2^]; mean ± SEM; *n*)	23.27 ± 0.43; 55	24.69 ± 0.66; 39	0.06
Sweat volume ([mL]; mean ± SEM; *n*)			
all	108.12 ± 8.59; 59	101.10 ± 12.27; 39	0.63
male	131.44 ± 9.65; 31	147.50 ± 20.57; 16	0.42
female	82.30 ± 13.15; 28	68.83 ± 11.16; 23	0.53

**Table 2 ijms-25-04658-t002:** Sweat pH and electrolyte concentrations in healthy controls and CholU patients.

	pH	Na^+^ [mM]	Cl^−^ [mM]	K^+^ [mM]	Ca^2+^ [mM]	Mg^2+^ [mM]
Healthy Controls						
MW ± SEM	7.33 ± 0.25	42.51 ± 3.13	41.47 ± 1.99	6.19 ± 0.61	1.04 ± 0.11	0.27 ± 0.06
*n*	20	39	39	39	39	25
CholU patients						
MW ± SEM	7.58 ± 0.21	47.60 ± 3.34	45.58 ± 2.23	8.54 ± 0.57	1.40 ± 0.10	0.27 ± 0.04
*n*	20	36	36	36	36	25
*p*-value	0.46	0.27	0.17	**0.01**	**0.02**	0.95

## Data Availability

Data is contained within the article and Appendix A.

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
