# Peer review of "Altered Sweat Composition Due to Changes in Tight Junction Expression of Sweat Glands in Cholinergic Urticaria Patients"

_ijms, 2024, doi:10.3390/ijms25094658_

Round 1
Reviewer 1 Report
Comments and Suggestions for Authors
The authors investigated the sweating abnormality of patients with cholinergic urticaria. They collected sweat from patients and healthy individuals and obtained skin biopsies. The concentration of K+ was increased and immunostainings of claudin-3 and claudin-10b were altered in patients. The results are interesting and valuable because they were obtained from clinical samples. The sweat provocation model in isolated skin is also an interesting part of the report.
I have several minor comments:
1. Please avoid using abbreviations without explanation. “CholU” should be spelled out in the title, also FFPE and TJ in the abstract.
2. Was there any correlation with age?
3. Indicate in Figures (legends) 2, 5 and 7 how many biological replicates were investigated. Are representatives images shown? How much variability was observed?
Reviewer 2 Report
Comments and Suggestions for Authors
1. How were CholU diagnosed in patients ?
2. Dermcidin staining interpretation (Figure 1) : how can you conclude that a decrease of dermcidin staining corresponds to a decrease of whole sebum secretion ? It might be possible that the production of dermcidin is decreased without a decrease of the volume of sebum secretion.
3. Figures 1 and 2 : without a counter-staining the pictures are difficult to interpret. The differences between patients and controls appear not significant.
4. Figure 6a : if carbachol was used to stimulate sweat secretion, then replace "Acetylcholine" by "Carbachol".
5. Replace "dermicidin" by "dermcidin" everywhere in the text.
6. Finally, the data are not really convincing. In the discussion several hypotheses and possible mechanisms are described, but they are not supported by the data.
Reviewer 3 Report
Comments and Suggestions for Authors
This is unique study.Well-wrtten and doccumented.
1.It will be nice to have western blot analysis of expression od TJ proteins
2.add study limitations
Round 2
Reviewer 2 Report
Comments and Suggestions for Authors
The authors made several changes in the text and added new data (Figure 2d-f), rendering the paper more understandable.
The field of Figure 1 is too small to ensure that there is no leakage of dermcidin in surrounding sebaceous glands. The Figure 1 is so not really convincing.